# BRAIN SIGNAL RENDERING: UNIFYING EEG VIDEO REPRESENTATIONS FOR SUBJECT-LEVEL FEW-SHOT LEARNING

## ABSTRACT

EEG modeling faces two core challenges: nonlinear, non-stationary dynamics and severe channel mismatch across datasets. We introduce Brain Signal Rendering (BSR), a new paradigm that reframes EEG representation learning as a rendering problem. BSR transforms EEG spectrograms into spatialized dynamic 'EEG videos', making representations invariant to electrode layouts and sampling protocols while preserving neural topology. Building on this, we propose EEG Consolidation — a unified multi-task training paradigm that integrates heterogeneous EEG-video data to adapt models to EEG-specific dynamics, improve data efficiency, reduce overfitting, and boost cross-task generalization. Crucially, BSR with EEG Consolidation enables subject-level few-shot learning, where each subject is treated as a distinct task requiring adaptation from minimal data. We validate this setting as a realistic benchmark and demonstrate substantial performance gains, establishing a scalable and interpretable framework toward foundation models for brain signals.

## 1 INTRODUCTION

Electroencephalography (EEG) offers one of the richest and most accessible windows into brain activity, driving advances in seizure detection (Shoeb & Guttag, 2010; Chen et al., 2025; Tegon et al., 2025), motor imagery (Ma et al., 2022), and emotion recognition (Duan et al., 2013; Zheng & Lu, 2015). Despite decades of progress, two fundamental barriers persist: (i) EEG signals are inherently *nonlinear* and *non-stationary*, making their spatiotemporal dynamics difficult to capture; (ii) electrode layouts vary widely across datasets, resulting in severe channel mismatch that impedes cross-domain generalization.

Recent deep learning advances, from task-specific networks (Jing et al., 2023) to large-scale foundation models (Yang et al., 2023; Jiang et al., 2024b; Wang et al., 2024a;b), have improved EEG representation learning significantly. Yet these models largely retain rigid, channel-first architectures that overlook a core reality of EEG: channels are not independent features, but samples from a *spatially structured* sensor array. This limitation hinders their ability to adapt in few-shot settings, especially under channel heterogeneity, making existing large-scale evaluation protocols insufficient for real-world EEG deployment.

**A New Perspective: EEG as a Physical Projection.** We depart from this channel-first paradigm by reinterpreting EEG not as a flat vector, but as the output of a *physical measurement process*. Electrodes form a two-dimensional sensor array that projects latent neural dynamics in three spatial dimensions plus time. From this viewpoint, channel mismatch is not noise but a change in perspective — analogous to how multiple cameras capture different projections of the same scene. This reframing transforms the objective of EEG representation learning: *from directly learning task-specific embeddings to inverting the projection and recovering the underlying spatiotemporal neural dynamics.*

**Brain Signal Rendering (BSR).** Motivated by the insight that EEG should be treated as a physical measurement process, we propose Brain Signal Rendering (BSR), a novel framework bridging raw EEG signals and powerful video foundation models. BSR treats EEG spectrograms as structured projections of a latent neural field and transforms them into a physically grounded visual format: a

Figure 1: Brain Signal Rendering (BSR) framework, which spatializes EEG spectrograms into dynamic "EEG videos".

dynamic image sequence or 'EEG video'. This is achieved by spatializing electrodes according to their physical coordinates, preserving the topology of neural activity as illustrated in Figure 1. The resulting representation encodes both spectral content and electrode geometry, making it directly compatible with video foundation models such as VideoMAE (Tong et al., 2022), whose spatiotemporal inductive biases align naturally with neural dynamics.

By decoupling representation learning from task-specific classification, BSR enables robust generalization across datasets and rapid adaptation to new conditions. Building on this, we introduce **EEG Consolidation** — a consolidated multi-task training paradigm that integrates heterogeneous EEG-video data. EEG spatialization renders representations invariant to electrode layouts and sampling protocols, enabling diverse datasets to be unified for joint training. This consolidation not only improves data efficiency and accelerates learning, but also reduces overfitting and markedly boosts generalization across tasks, paving the way toward scalable, adaptable EEG representation learning.

Our BSR framework, together with EEG Consolidation, enables *subject-level few-shot learning*. A key challenge in real-world EEG applications is adapting to new subjects, where variability in acquisition hardware, protocols, and individual physiology severely limits generalization. To rigorously test this ability, we introduce *subject-level few-shot learning* as our main experimental benchmark. Here, each subject is treated as a distinct task, requiring adaptation with only a few recorded sessions. This setting directly evaluates model adaptability in realistic deployment scenarios, and we demonstrate that BSR combined with multi-task EEG Consolidation delivers substantial performance gains under this demanding regime.

**Contributions.** This work makes four key contributions. (1) We introduce **Brain Signal Rendering (BSR)**, a physics-informed framework that reframes EEG modeling as a rendering problem and transforms raw signals into spatiotemporal video representations suitable for video foundation models such as VideoMAE. (2) We propose **EEG Consolidation**, a multi-task fine-tuning paradigm that integrates heterogeneous EEG-video data to update VideoMAE. This process adapts VideoMAE to the unique spatiotemporal characteristics of EEG data, enabling improved cross-task representation learning and robustness. (3) We define **Subject-level Few-shot Learning**, a new benchmark that evaluates subject adaptation by treating each individual as a distinct task and requiring models to adapt with only a few calibration sessions. (4) Through extensive experiments across multiple datasets, we show that BSR consistently outperforms prior EEG representation learning methods, establishing a scalable, interpretable, and data-efficient foundation for EEG modeling.

## 2 RELATED WORKS

**Deep Models for EEG Data**. Recent advances in deep EEG modeling have explored various architectures for cross-dataset generalization and task adaptability. BIOT (Yang et al., 2023) segments EEG into fixed-duration patches per channel and employs independent temporal and spatial embeddings to enable cross-data pre-training. LaBraM (Jiang et al., 2024b) extends this approach by incorporating a neural tokenizer and large-scale pretraining, achieving notable performance gains.

CBraMod (Wang et al., 2024b) and EEGPT (Wang et al., 2024a) further demonstrate the effectiveness of deep architectures in single-task fine-tuning scenarios. More recent works, such as NeuroLM (Jiang et al., 2024a) and UniMind (Lu et al., 2025), advance toward multi-task EEG decoding, underscoring the growing interest in unified EEG modeling.

**Few-shot Learning for EEG Foundation Models.** Benchmarking EEG foundation models is an evolving field of research, with several few-shot learning paradigms recently proposed. AdaBrain-Bench (Wu et al., 2025), for instance, introduces a few-shot evaluation protocol that utilizes a fixed proportion of the training data. Similarly, BrainWave (Yuan et al., 2024) explores few-shot learning by evaluating models under 3-shot and 8-shot settings across tasks. In contrast to these approaches, we propose a *subject-level few-shot learning* paradigm, where pretrained models are required to generalize to *unseen tasks and channel configurations* during pre-training. This setup rigorously tests their generalization capabilities under extreme and realistic conditions. Building upon this rigorous testbed, our future efforts will incorporate the few-shot paradigm to address generalization to entirely unseen subjects, aligning with promising research directions such as Bhosale et al. (2022).

**VideoMAE for Few-shot Learning** VideoMAE (Tong et al., 2022) represents a breakthrough in self-supervised video representation learning, leveraging large-scale unlabeled video data to learn powerful, generalizable features that excel in few-shot settings. Its masked autoencoding paradigm enables the model to capture rich spatiotemporal dependencies efficiently, making it highly robust for cross-domain generalization. For example, Hatano et al. (Hatano et al., 2024) show that VideoMAE achieves significant gains in cross-domain few-shot action recognition by training separate models on multiple modalities and optimizing for domain-invariant features. Samarasinghe et al. (Samarasinghe et al., 2023) demonstrate that a VideoMAE-pretrained universal encoder can transfer effectively to unseen domains in few-shot video understanding tasks. We adopt VideoMAE as our backbone because its design naturally aligns with Brain Signal Rendering (BSR), which converts EEG into spatiotemporal "video" sequences encoding spectral and spatial neural dynamics. VideoMAE's strength in capturing rich spatiotemporal patterns and its efficiency in low-data regimes make it ideal for EEG videos. Combined with EEG Consolidation, this synergy forms a unified framework for robust subject-level few-shot EEG learning. Additionally, we also note that our framework is compatible with other recent video foundation models beyond VideoMAE; exploring such extensions lies beyond the scope of this work and is orthogonal to our core contributions.

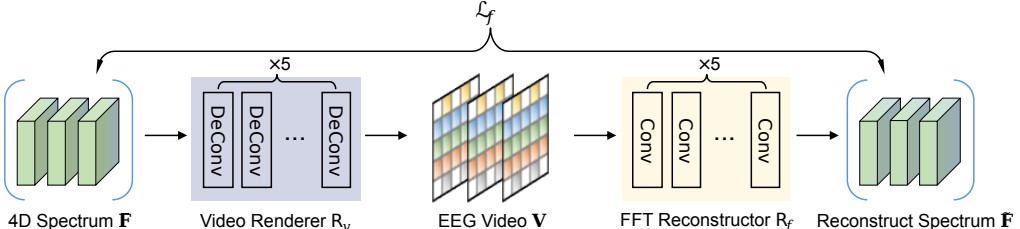

Figure 2: Framework of the BSR Render-Reconstruct pipeline.

# 3 METHODS

## 3.1 BRAIN SIGNAL RENDERING

**Task Definition.** Given a raw EEG sample represented as $\boldsymbol{X} \in \mathbb{R}^{c \times l}$, where $c$ denotes the number of electrode channels and $l$ denotes the number of sample timestamps, we aim to predict the task-specific one-hot label $\boldsymbol{y} \in \mathbb{N}^m$, where m represents the number of classes for the downstream EEG task.

**Spatial-Temporal Spectrum Preprocessing.** Firstly, we preprocess the raw EEG data separately along its temporal and spatial dimensions. To capture the time-varying frequency content of EEG signals, we apply the Short-Time Fourier Transform (STFT), denoted as $\text{stft}(\cdot)$, which decomposes the signal into a sequence of frequency spectra over time. This operation extracts both frequency and amplitude information from the inherently non-stationary EEG data, thereby enhancing its temporal-frequency representation. We then compute the magnitude ($\text{abs}(\cdot)$) of the resulting complex spec-

trogram to obtain the final temporal feature map $\mathbf{F}$:

$$\mathbf{F} = \mathrm{abs}\left(\mathrm{stft}(\boldsymbol{X}; t, d)\right), \quad \mathbf{F} \in \mathbb{R}^{n \times c \times f}, \tag{1}$$

where $t$ and $d$ denote the STFT window size and hop length, respectively; $n = 1 + \left\lfloor \frac{l-t}{d} \right\rfloor$ is the number of time windows, $c$ is the number of EEG channels, and $f$ is the number of frequency bins.

While $\mathbf{F}$ contains rich time–frequency information, it lacks explicit spatial encoding. Since each channel in $\mathbf{F}$ corresponds to an EEG electrode with known spatial coordinates on the scalp, we exploit this inherent spatial structure to embed positional information through channel rearrangement. To formalize this process, we define a user-specified channel spatialization map matrix $\boldsymbol{S} \in \mathbb{N}^{h \times w}$, where each element specifies the target spatial location for the corresponding channel. For instance, if $\boldsymbol{S}[1, 4] = 4$ and $\mathbf{F}[:, 4]$ corresponds to electrode FP2, this indicates that the FP2 spectrum should be rendered at the fourth patch in the first row of the spatial map, as illustrated in Figure 1. We then apply a brain signal spatialization algorithm to transform $\mathbf{F}$ into a spatially organized representation:

$$\overline{\mathbf{F}} = \mathcal{S}(\mathbf{F}, \boldsymbol{S}), \quad \overline{\mathbf{F}} \in \mathbb{R}^{n \times h \times w \times f}, \tag{2}$$

where $\mathcal{S}(\cdot)$ denotes the spatialization operation, and $\overline{\mathbf{F}}$ is a structured spatiotemporal EEG feature map suitable for subsequent multimodal processing.

---

**Algorithm 1** Brain Signal Spatialization $\mathcal{S}$

---

1: **Input:** fourier amplitude spectrum $\mathbf{F} \in \mathbb{R}^{n \times c \times f}$, spatialization map $\boldsymbol{S} \in \mathbb{N}^{h \times w}$
2: Initialize a spatialized 4D spectrum tensor $\overline{\mathbf{F}}$ of size $n \times h \times w \times f$
3: **for** $i = 1$ to $h$ **do**
4:     **for** $j = 1$ to $w$ **do**
5:         **if** $\boldsymbol{S}[i, j] > 0$ **then**
6:             $\overline{\mathbf{F}}[:, i, j] \leftarrow \mathbf{F}\left[:, \boldsymbol{S}[i, j]\right]$
7:         **else**
8:             $\overline{\mathbf{F}}[:, i, j] \leftarrow \mathbf{0}^{t \times n}$
9:         **end if**
10:     **end for**
11: **end for**
12: **Return** $\overline{\mathbf{F}}$

---

**Rendering Process.** After obtaining the 4D Fourier frequency map $\overline{\mathbf{F}} \in \mathbb{R}^{n \times h \times w \times f}$, we transform it into a structured EEG video representation

$$\mathbf{V} \in \mathbb{R}^{n \times H \times W \times 3}$$

using our *brain signal renderer* $\mathrm{R}_v$, as formalized in Equation (3). The renderer $\mathrm{R}_v$ comprises a sequence of cascaded deconvolution (transposed convolution) layers with equal kernel size and stride, which preserves the time–frequency content while mapping the spatialized EEG features into a dense RGB representation. Specifically, given the spatialized spectrum corresponding to a single time window $\overline{\mathbf{F}}_i \in \mathbb{R}^{h \times w \times f}$, the renderer produces an image-shaped tensor $\mathbf{V}_i \in \mathbb{R}^{H \times W \times 3}$:

$$\mathbf{V} = \left[\mathbf{V}_1, \mathbf{V}_2, \ldots, \mathbf{V}_n\right]^\top, \quad \mathbf{V}_i = \mathrm{R}_v(\overline{\mathbf{F}}_i), \quad i = 1, 2, \ldots, n, \tag{3}$$

where $H$ and $W$ are hyperparameters of the renderer defining the spatial resolution of each frame, and $n$ denotes the number of time windows. The resulting tensor $\mathbf{V}$ constitutes a spatiotemporal sequence, effectively an *EEG video*, which preserves both spectral and spatial information for downstream video-based processing.

**Reconstruction Process.** To train the renderer $\mathrm{R}_v$, we jointly learn a *reconstructor* $\mathrm{R}_f$ that inverts the rendering process by reconstructing the spatialized Fourier frequency map $\overline{\mathbf{F}}_i$, as formalized in Figure 2. The reconstructor produces $\tilde{\mathbf{F}}_i \in \mathbb{R}^{h \times w \times f}$, as formalized in Equation (4):

$$\tilde{\mathbf{F}} = \left[\tilde{\mathbf{F}}_1, \tilde{\mathbf{F}}_2, \ldots, \tilde{\mathbf{F}}_n\right]^\top, \quad \tilde{\mathbf{F}}_i = \mathrm{R}_f(\mathbf{V}_i), \quad i = 1, 2, \ldots, n, \tag{4}$$

where $\tilde{\mathbf{F}} \in \mathbb{R}^{n \times h \times w \times f}$ is the reconstructed spatialized feature map. The reconstructor $\mathrm{R}_f$ adopts a symmetrical architecture to the renderer, replacing each deconvolution (transposed convolution)

layer with a corresponding convolution layer, while maintaining identical kernel dimensions, stride, and layer depth. This symmetry ensures effective inversion of the rendering process while preserving spectral and spatial information. Implementations of the renderer and reconstructor used in this study are detailed in Table 3.

The entire system is trained end-to-end with an L1 reconstruction loss, defined as:

$$\mathcal{L}_f = \frac{1}{N} \sum_{i=1}^{N} \left| \bar{\mathbf{F}} - \tilde{\mathbf{F}} \right|, \quad N = n \times h \times w \times f, \tag{5}$$

where $N$ is the total number of elements in $\bar{\mathbf{F}}$ and $\tilde{\mathbf{F}}$, ensuring the loss measures the element-wise absolute error over the entire spatiotemporal frequency representation.

## 3.2 FINE-TUNING VIDEOMAE WITH EEG VIDEOS

The rendered output $\mathbf{V} \in \mathbb{R}^{n \times H \times W \times 3}$ possesses the same spatiotemporal properties as ordinary video inputs. Since no numerical range constraints are imposed during the rendering stage, we normalize each frame $\mathbf{V}_i$ using Contrast Limited Adaptive Histogram Equalization (CL-AHE), denoted as $\mathrm{T}(\cdot)$, to obtain the final video representation:

$$\hat{\mathbf{V}}_i = \mathrm{T}(\mathbf{V}_i), \quad i = 1, 2, \ldots, n. \tag{6}$$

This normalization ensures consistent intensity distribution across frames, enhancing the stability and performance of subsequent video-based processing. We then leverage a pre-trained video foundation encoder and fine-tune it for various downstream EEG tasks.

In this work, we adopt **VideoMAE** (Tong et al., 2022) as our pre-trained video encoder $\mathcal{V}$, motivated by its strong capability to capture spatiotemporal patterns through masked autoencoding and its superior generalization performance. Following the default VideoMAE setup, we append a linear layer on top of the average-pooled hidden states as the task-specific head $\mathcal{H}$. Before being fed into VideoMAE, all rendered EEG videos $v$ are resized to a resolution of $(224, 224)$ and temporally sampled to 16 frames, ensuring compatibility with the pre-trained encoder and enabling efficient fine-tuning.

**Updating BSR-VideoMAE via EEG Consolidation.** Prior EEG modeling often fine-tunes models separately for each dataset, limiting shared knowledge and increasing training cost. Our EEG-to-video rendering produces representations invariant to timestamps and electrode layouts, enabling integration of heterogeneous datasets into a consolidated training paradigm.

On the other hand, VideoMAE, designed for natural videos, cannot optimally handle EEG-video data without adaptation. To address this, we propose *EEG Consolidation* — a multi-task fine-tuning strategy that unifies diverse EEG-video data to update BSR-VideoMAE, aligning it with EEG-specific spatiotemporal dynamics.

EEG Consolidation leverages common patterns across tasks and complementary information from multiple datasets, improving efficiency, robustness, and generalization. This approach not only reduces overfitting and accelerates learning but also enables VideoMAE to fully exploit the potential of our rendering-based EEG-to-video framework across diverse EEG tasks. Specifically, suppose there are $M$ tasks. For each task $m$, the corresponding dataset is mapped through a unified pre-trained video encoder $\mathcal{V}$ and a task-specific head $\mathcal{H}_m$ to obtain the final classification logits via composing functions,

$$\tilde{\boldsymbol{y}}_m = \mathcal{H}_m \circ \mathcal{V}(\hat{\mathbf{V}}_m), \quad m = 1, 2, \ldots, M. \tag{7}$$

The multiple tasks are learned jointly by optimizing the aggregated multi-task loss:

$$\min_{\theta_{\mathcal{V}}, \{\theta_{\mathcal{H}_m}\}_{m=1}^{M}} \sum_{m=1}^{M} \lambda_m \cdot \mathcal{L}^{(m)}(\tilde{\boldsymbol{y}}_m, \boldsymbol{y}_m), \tag{8}$$

where $\mathcal{L}^{(m)}$ denotes the task-specific loss function and $\lambda_m$ is a binary indicator:

$$\lambda_m = \begin{cases} 1, & \text{if any data } \boldsymbol{X}_m \text{ exists in the current training batch,} \\ 0, & \text{otherwise.} \end{cases}$$

In this study, we use the cross-entropy loss for all $\mathcal{L}^{(m)}$.

**Discussion.** Our BSR-VideoMAE, empowered by *EEG Consolidation*, demonstrates strong potential as a universal EEG foundation model. By transforming EEG into video-like representations, it opens the door to transfer powerful capabilities from video models to EEG tasks. Early scaling experiments show promising results (see Appendix E), but fully realizing this vision requires consolidating vastly more EEG-video data and substantial computational resources. We view this as a key future direction and invite the community to contribute to advancing BSR-VideoMAE toward a truly generalizable EEG foundation model.

### 3.3 SUBJECT-LEVEL FEW-SHOT LEARNING

A key challenge in EEG analysis is the substantial variability across subjects, arising from differences in acquisition equipment, sampling protocols, and individual neurophysiological characteristics. This inter-subject variability often leads to poor generalization of models trained on existing datasets when applied to new individuals, thereby limiting the practical applicability of EEG-based systems in real-world scenarios.

The motivation for *subject-level few-shot learning* is to explicitly evaluate and improve a model's ability to adapt to new subjects using minimal labeled data. This setting reflects realistic application scenarios, such as personalized brain-computer interfaces, where collecting extensive labeled EEG data for every new user is impractical.

To this end, we propose a novel benchmark task called *subject-level few-shot learning*, where each subject is treated as a distinct task. For a new subject $s$, we treat all sampled data from that subject as the subject-specific dataset $\mathcal{D}_s = \{(\boldsymbol{X}_{s,j}, \boldsymbol{y}_{s,j})\}_{j=1}^{N_s}$, where $N_s$ denotes the total number of samples available. We divide $\mathcal{D}_s$ into a small training subset $\mathcal{D}_s^{\text{train}}$ and a testing subset $\mathcal{D}_s^{\text{test}}$, with $|\mathcal{D}_s^{\text{train}}| \ll |\mathcal{D}_s|$.

The objective is to fine-tune the pre-trained VideoMAE model using only $\mathcal{D}_s^{\text{train}}$, and then evaluate its performance on $\mathcal{D}_s^{\text{test}}$:

$$\min_{\theta_{\mathcal{V}}, \theta_{\mathcal{H}_s}} \mathcal{L}^{(s)}\big(\mathcal{H}_s \circ \mathcal{V}(\mathcal{R}(\mathcal{D}_s^{\text{train}})), \, \boldsymbol{y}_s\big), \tag{9}$$

where $\mathcal{L}^{(s)}$ denotes the loss function for subject $s$ (e.g., cross-entropy), and $\mathcal{R}$ denotes the EEG Video rendering process.

By focusing on rapid adaptation to unseen subjects with only a few samples, *subject-level few-shot learning* provides a realistic and rigorous measure of the generalization ability of EEG video models, and demonstrates the practical advantage of our rendering-based EEG-to-video framework combined with VideoMAE fine-tuning.

## 4 EXPERIMENTS

### 4.1 EVALUATION DATASETS

For pre-training VideoMAE via EEG Consolidation and for comparisons with baseline methods, we use two EEG datasets. The **TUAB** dataset Obeid & Picone (2016) is designed for abnormal detection and consists of two categories: normal and abnormal. The **TUEV** dataset Obeid & Picone (2016) is an event classification benchmark with six categories, namely spike and sharp wave (SPSW), generalized periodic epileptiform discharges (GPED), periodic lateralized epileptiform discharges (PLED), eye movement (EYEM), artifact (ARTF), and background (BCKG).

For subject-level few-shot fine-tuning, we use four EEG datasets. The **SEED** dataset (Duan et al., 2013; Zheng & Lu, 2015) targets emotion classification with three categories (negative, neutral, positive), and each subject has three sessions, split into train:validation:test of 1:1:1. The **SEED-VII** dataset (Jiang et al., 2025) extends this to seven emotion categories (happy, surprise, neutral, sad, disgust, fear, anger); each subject has four sessions, but as no single session covers all categories, we used a 2:2 train:test split (sessions 1 and 3 for training, sessions 2 and 4 for testing). The **SHU-MI** dataset (Ma et al., 2022) is a large-scale motor imagery dataset with two classes (left-hand, right-hand), also split 1:1:1 across three sessions. Finally, the well-known **BCICIV-2a** dataset (Brunner

et al., 2008) focuses on motor imagery with four classes (left-hand, right-hand, both feet, tongue), where we adopt the official 1:1 train:test split. In addition to these, we use the large-scale **TUEG** dataset Obeid & Picone (2016), containing 26,846 clinical EEG recordings collected from 2002 to 2017, to pre-train the BSR renderer. Detailed information and preprocessing pipeline about these datasets are summarized Appendix C

## 4.2 EXPERIMENTAL SETUP

**Evaluation Metrics.** We evaluate model performance using a set of metrics tailored to each task. For **binary classification**, we report balanced accuracy (**B-Acc.**), area under the receiver operating characteristic curve (**AUROC**), and area under the precision-recall curve (**AU-PR**), where AU-PR is particularly robust for imbalanced datasets by focusing on the positive class. For **multi-class classification**, we report balanced accuracy, **Cohen's Kappa** ($\kappa$), which adjusts for chance agreement between predictions and labels, and the **weighted F1 score (F1w)**, the harmonic mean of precision and recall weighted by class sample sizes.

**Experiment Platform.** All experiments were conducted on a machine with $8 \times$ NVIDIA H100-80G GPUs, an Intel Xeon Gold 6330 CPU, and 200 GB RAM, using Python3.11.11, PyTorch2.5.1, and CUDA12.2. Video I/O was implemented with OpenCV-Python and PIL.

**Baselines.** To evaluate BSR, we compared against five FFT-based baselines. **FFCL** (Li et al., 2022) uses a CNN-LSTM fusion network for motor imagery classification, combining spatial and temporal features. **ContraWR** (Yang et al., 2021) applies self-supervised learning to improve sleep staging by leveraging unlabeled EEG data. **CNN-Transformer** (Peh et al., 2022) employs a CNN-Transformer hybrid with belief matching loss for multi-type EEG artifact detection, maximizing artifact rejection while preserving clean signals. **BIOT** (Yang et al., 2023) presents a flexible biosignal encoder for multi-dataset pre-training and task-specific fine-tuning across diverse EEG formats. **LaBraM** (Jiang et al., 2024b) proposes a unified EEG foundation model to address the limitations of specialized deep learning approaches.

## 4.3 PRE-TRAINING SETTINGS AND RESULTS

For the BSR framework, both the Video Renderer and VideoMAE require pretraining, with structural hyperparameters detailed in Table 3. The renderer was unsupervisedly pretrained on the TUEG (Obeid & Picone, 2016) dataset for 200 epochs using the Adam optimizer with a learning rate of $1 \times 10^{-5}$. For VideoMAE pretraining, we loaded weights from Kinetics-400 (Kay et al., 2017) and jointly fine-tuned on TUAB and TUEV (Obeid & Picone, 2016) for 10 epochs using the AdamW optimizer (learning rate $1 \times 10^{-5}$, weight decay $1 \times 10^{-4}$) with a cosine annealing scheduler. To prevent potential data leakage from overlap between LaBraM's official pretraining dataset and our few-shot sets, we did not use the official LaBraM-base weights, and instead pretrained LaBraM-base separately on TUAB and TUEV for 50 epochs using the model's recommended hyperparameters and official vqnsp weights.

During testing, we observed that the trained renderer is highly robust. Even when the test data is subjected to various random noise disturbances, the quality of the rendered videos remains consistent. This provides stable input features for subsequent few-shot tuning and demonstrates significant practical value for real-world applications.

## 4.4 SUBJECT-LEVEL FEW-SHOT FINE-TUNING

In this experiment, we show that BSR-VideoMAE sets a new state-of-the-art in few-shot EEG-video learning, outperforming all baselines and demonstrating unmatched robustness across datasets.

For both BSR-VideoMAE and LaBraM, we initialized model weights from pretraining on the TUAB and TUEV datasets. In contrast, all other baseline methods were trained from scratch, which places them at a disadvantage in leveraging prior knowledge. A key advantage of BSR-VideoMAE is its consistent input size of $224 \times 224 \times 3$ for all experiments, ensuring uniform processing and robustness. Other methods rely on variable input channel configurations determined by the number of electrodes in each dataset, introducing additional variability and potential optimization challenges. Since each subject represents an independent dataset, we report the average performance across all

Table 1: Subject-level few-shot learning results for the emotion classification task

| Methods | Pretrain | SEED | | | SEED-VII | | |
|---|---|---|---|---|---|---|---|
| | | B-Acc. | $\kappa$ | F1w | B-Acc. | $\kappa$ | F1w |
| FFCL | — | 0.4100 | 0.1103 | 0.3570 | 0.1682 | 0.0280 | 0.1216 |
| ContraWR | — | 0.3589 | 0.0353 | 0.2124 | 0.1569 | 0.0237 | 0.0976 |
| CNN-Transformer | — | 0.4421 | 0.1594 | 0.3322 | 0.1629 | 0.0204 | 0.0938 |
| BIOT | — | 0.3677 | 0.0511 | 0.2245 | 0.1928 | 0.0524 | 0.1303 |
| LaBraM | TUAB+EV | 0.3932 | 0.0864 | 0.3342 | 0.1428 | 0.0000 | 0.0203 |
| BSR-VMAE (ours) | TUAB+EV | **0.4800** | **0.2169** | **0.4500** | **0.1948** | **0.0558** | **0.1559** |

Table 2: Subject-level few-shot learning results for the motor imagery task

| Methods | Pretrain | SHU-MI | | | BCICIV-2a | | |
|---|---|---|---|---|---|---|---|
| | | B-Acc. | AUROC | AU-PR | B-Acc. | $\kappa$ | F1w |
| FFCL | — | 0.5271 | 0.5608 | 0.5644 | 0.2935 | 0.0581 | 0.2673 |
| ContraWR | — | 0.5105 | 0.5886 | 0.5962 | 0.2843 | 0.0458 | 0.2505 |
| CNN-Transformer | — | 0.5388 | 0.5988 | 0.5992 | 0.2531 | 0.0041 | 0.1132 |
| BIOT | — | 0.5456 | 0.5790 | 0.5828 | 0.2735 | 0.0314 | 0.1931 |
| LaBraM | TUAB+EV | 0.5288 | 0.5535 | 0.5568 | 0.2650 | 0.0201 | 0.1802 |
| BSR-VMAE (ours) | TUAB+EV | **0.5808** | **0.6144** | **0.6247** | **0.3029** | **0.0705** | **0.2821** |

subjects to ensure fair and comprehensive evaluation. Tables 1 and 2 summarize the results for emotion classification and motor imagery tasks, respectively, with the best performance in bold and the second-best underlined.

Across all few-shot experiments, BSR-VideoMAE consistently outperforms competing methods, establishing it as an effective model for EEG-video representation learning. Notably, the improvement is particularly pronounced on the SEED and SHU-MI datasets, which likely have fewer categories and thus a relatively easier classification space. Some baseline methods fail to converge in these settings, resulting in zero $\kappa$ scores, indicating that the limited data in few-shot tasks is insufficient for reliable training without a robust pretrained model. These results demonstrate that BSR-VideoMAE's transformer-based architecture, combined with EEG-video pretraining and consistent input processing, delivers superior generalization and stability across diverse datasets and tasks.

### 4.5 ABLATION STUDY ON PRE-TRAINING CHANNELS

While BSR-VideoMAE has demonstrated outstanding generalization ability across various unseen channel settings, the precise source of this performance gain remains unclear. To address this, we conduct an ablation study on channel utilization to pinpoint the factors contributing to the improved generalization. Specifically, we evaluate the subject-level few-shot learning performance using three distinct channel rendering strategies: (1) all channels are rendered, which is the same with previous experiments, (2) channels contained in the pretraining are masked and other are kept, (3) only channels used for pretraining are kept. The visualization detailing these three channel strategies is provided in Appendix 6. Under each defined channel strategy, the performance is assessed using two models: the Baseline model, which is the Kinetics-400 initialized VideoMAE; and the pre-trained model, derived from the baseline after further EEG-Consolidation on the TUAB and TUEV datasets.

The results of the channel-based ablation study are detailed in Figure 3. Generally, the Pre-trained (PT) models, which leveraged EEG Consolidation (pre-training on TUAB and TUEV), consistently demonstrated performance improvement across the majority of experimental settings compared to the Baseline (BT) models. Crucially, this performance gain was also observed in most of the w/o PT-Ch groups, where channels used during pre-training were deliberately masked out. This finding strongly suggests that the pre-training mechanism induces a robust and abstract EEG representation within the BSR-VideoMAE model, rather than merely memorizing channel-specific patterns.

### 4.6 SCALING EXPERIMENT ON DATASET SIZE AND MODEL INITIALIZATION

To rigorously evaluate the collective effects of model initialization and pre-training data volume on performance scaling, we test two distinct initialization strategies: Load Kinetics, where the model

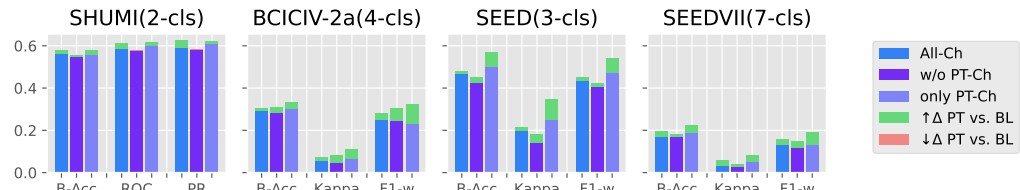

Figure 3: Ablation study on the effect of pre-training channels. The performance metrics are shown for different channel strategies: all channels (All-Ch), channels which are not contained in pre-training (w/o PT-Ch), and only pre-training channels (only PT-Ch). BT (Baseline) uses Kinetics-400 weights, while PT (Pre-trained) models are further pre-trained on TUAB and TUEV.

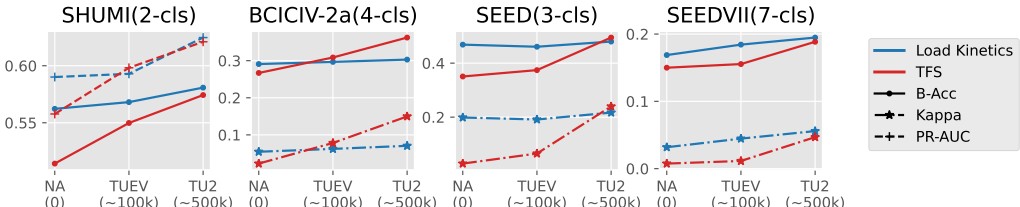

Figure 4: Scaling experiment on pre-training dataset size and model initialization. The results compare models initialized via Load Kinetics (further pre-trained from Kinetics-400 weights) versus those Trained From Scratch (TFS) . The X-axis indicates the pre-training dataset used (approximate number of samples), where TU2 denotes the combined TUAB and TUEV dataset.

inherits weights from Kinetics-400 pre-training, and Trained From Scratch, where the model is randomly initialized. We consider three tiers of pre-training data settings: No Pre-training (NA), pre-training solely on the TUEV dataset, and EEG Consolidation (TU2), which involves pre-training on the combined TUAB and TUEV datasets. These settings allow us to systematically assess the contribution of the initial general-domain knowledge versus domain-specific data volume to the final few-shot fine-tuning performance.

As the results presented in Figure 4 clearly demonstrate, the scaling experiment yields two significant observations. First, performance consistently increases as the volume of the EEG pre-training dataset grows, confirming the expected positive relationship between domain-specific data volume and model performance enhancement. Second, while the Load Kinetics initialization provides a substantial performance advantage in data-scarce scenarios, this gap narrows significantly as the pre-training data volume increases. This convergence suggests that the spatial-temporal representations learned by the BSR pipeline from extensive EEG data (TU2) eventually mitigates the influence of the initial bias. Crucially, this final performance similarity implies that the feature hierarchy captured by BSR-VideoMAE from raw EEG signals is intrinsically analogous to the fundamental feature structure learned by the base VideoMAE on natural videos, highlighting the effectiveness of our pipeline in adapting general video modeling principles to the EEG domain.

## 5 CONCLUSIONS AND FUTURE WORKS

We present **Brain Signal Rendering (BSR)**, a framework that transforms EEG signals into spatiotemporally structured videos, enabling unified multi-task learning and robust subject-level few-shot adaptation across heterogeneous datasets, tasks, and electrode configurations. By pairing EEG-to-video representations with pre-trained video encoders and **EEG Consolidation**, BSR achieves strong performance and generalization in diverse BCI tasks, reframing EEG modeling as a rendering problem rather than a channel-first learning task.

Looking ahead, two promising directions can further amplify this impact: (1) *Efficiency scaling*—exploring lighter or EEG-specialized encoders to reduce computational cost without sacrificing performance, and pre-training on larger scale combined EEG datasets; (2) *Rendering exploration*—investigating alternative transformation methods, and using metrics which could reflect brain activity such as band features for rendering evaluation. These avenues promise to advance BSR toward a scalable, adaptable foundation for EEG representation learning, paving the way for practical, high-performance brain-computer interface systems.

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

## A THE USE OF LARGE LANGUAGE MODELS

In this work, we only use LLMs for polish writing and related work discovery.

## B ADDITIONAL INFORMATION FOR BRAIN SIGNAL RENDERING

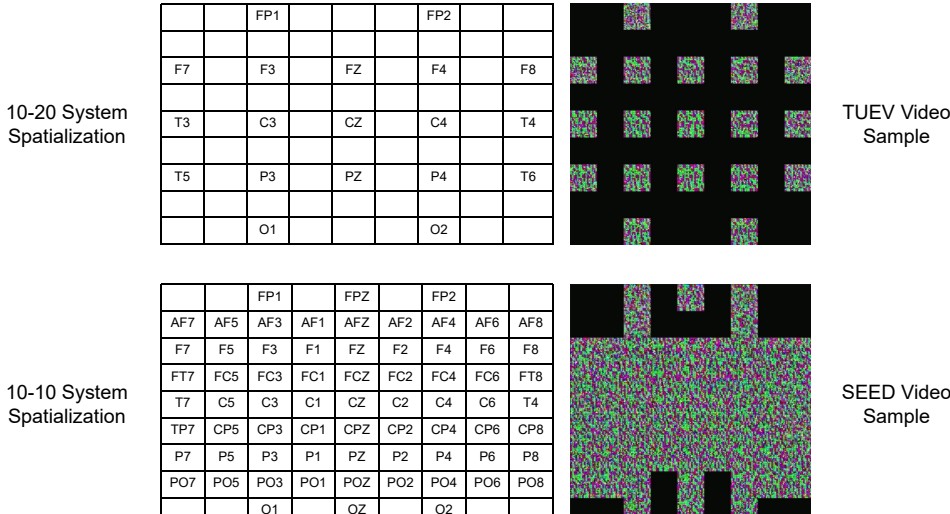

Figure 5: Spatialization matrix and EEG video examples.

Table 3: Hyperparameters for BSR Renderer and Reconstructor

| Layer | Renderer - ConvTranspose2d (in-ch, out-ch, kernel-size, stride) | Reconstructor - Conv2d (in-ch, out-ch, kernel-size, stride) |
|---|---|---|
| 1 | (101, 50, 2, 2) | (3, 6, 2, 2) |
| 2 | (50, 25, 2, 2) | (6, 12, 2, 2) |
| 3 | (25, 12, 2, 2) | (12, 25, 2, 2) |
| 4 | (12, 6, 2, 2) | (25, 50, 2, 2) |
| 5 | (6, 3, 2, 2) | (50, 101, 2, 2) |

Figure 5 illustrates the spatialization for both the 10-20 and 10-10 systems, along with video examples from the TUEV and SEED datasets. It is important to note that the BSR renderer's pre-training is independent of any specific EEG system. Once pre-trained, the renderer can be directly applied to any system defined by a user-specified spatialization matrix without requiring further training.

The code will be made publicly available upon paper acceptance.

## C DATASET INFORMATION AND PREPROCESSING

We preprocess the six datasets introduced in 4 as follows:

- For TUEG, TUAB and TUEV, we follow the pipeline of Jiang et al. (2024b), which employs a filter between 0.1Hz and 75Hz, as well as a notch filter of 60Hz.
- For SEED and SEED-VII, we directly use the official preprocessed data, which pipeline is introduced in Zheng & Lu (2015); Jiang et al. (2025)
- For SHUMI, we follow the pipeline of Wang et al. (2024b), which takes no preprocess except the down-sampling.
- For BCICIV-2a dataset, we employs a band filter from 0Hz to 38 Hz.

Table 4: Basic information of the datasets

| Datasets | # Channels | # Classes | Duration | # Samples | # Subjects (for Few-shot) |
|---|---|---|---|---|---|
| **Pre-training** | | | | | |
| TUAB | 19 | 2 | 10 seconds | 409455 | — |
| TUEV | 19 | 6 | 5 seconds | 113353 | — |
| **Subject-level Few-shot** | | | | | |
| SEED | 62 | 3 | 10 seconds | 13860 | 15 |
| SEED-VII | 62 | 7 | 10 seconds | 27340 | 20 |
| SHU-MI | 32 | 2 | 4 seconds | 11988 | 25 |
| BCICIV-2a | 22 | 4 | 4 seconds | 5184 | 9 |

All EEG data are down-sampled to 200 Hz and stored in **unipolar** form, which differs from some baselines such as BIOT (Yang et al., 2023).

## D    ADDITIONAL INFORMATION FOR SUBJECT-LEVEL FEW-SHOT

**Hyperparameters.**

- BSR-VideoMAE will be trained for 25 epochs for all experiments, with a learning rate using the AdamW optimizer with a learning rate of 1e-5, weight decay of 1e-4 and a cosine annealing scheduler.

- LaBraM is trained for 50 epochs with its recommended hyperparameters for all experiments.

- For other baselines, if validation set is applicable, they will be trained for 50 epochs with an early stop callback on AUROC (binary classification) or $\kappa$ (multiclass classification). Elsewere they will be trained for 15 epochs to prevent overfitting. Other hyperparameters are referred to (Yang et al., 2023).

To ensure the train-(val)-test split for all methods is strictly consistent, the rendering process on all evaluation datasets does not access the raw data but the processed EEG segments, which are the direct input of all baseline methods.

## E    FULL-DATASET FINE-TUNING

The BSR-VideoMAE model was trained jointly on the TUAB and TUEV datasets with the multi-task fine-tuning strategy, loaded with weights pretrained on Kinetics-400, and evaluated without further tuning on each dataset. For optimization, we used the AdamW optimizer with a learning rate of 1e-5 and a weight decay of 1e-4. The training process incorporated a cosine annealing scheduler and was performed with a gradient accumulation of 8. LaBraM was pre-trained for 50 epochs on the training partitions of the TUAB and TUEV datasets, followed by a separate 10-epoch fine-tuning stage on each. We adopted the hyperparameter settings detailed in (Jiang et al., 2024b). All BIOT models were trained from scratch to avoid channel mismatch, as the official weights were pre-trained with a bipolar electrode configuration. For the remaining baseline methods, we adhered to the hyperparameter settings of (Yang et al., 2023). All reported results represent the average performance over three runs with different random seeds.

**Dataset Split** Since the original TUAB and TUEV dataset already provides the split of training and test sets, we use 10% of the training set for validation.

Table 5 presents the fine-tuning results on the TUAB and TUEV datasets. The multi-task fine-tuning was designed to enable BSR-VideoMAE to learn robust representations for few-shot learning, and these full-dataset results serve as a baseline for comparison. One limitation may be the use of a model pretrained on natural videos rather than EEG rendered videos, as mentioned in Figure 3. Despite the challenges introduced by the multi-task setting and the natural video pre-training, BSR-VideoMAE still achieved near state-of-the-art performance on TUEV and a competitive result on TUAB, demonstrating the scalability and adaptability of the EEG Consolidation paradigm.

Table 5: Fine-tuning on TUAB and TUEV

| Methods | TUAB | | | TUEV | | |
|---|---|---|---|---|---|---|
| | B-Acc. | AUROC | AU-PR | B-Acc. | $\kappa$ | F1w |
| FFCL | 0.7510 | 0.8617 | 0.8718 | 0.3757 | 0.3273 | 0.6603 |
| ContraWR | 0.7746 | 0.8637 | 0.8711 | 0.3833 | 0.3510 | 0.6706 |
| CNN-Transformer | 0.7674 | **0.8796** | **0.8847** | 0.3360 | 0.3355 | 0.6646 |
| BIOT | 0.7858 | 0.8724 | 0.8768 | 0.4145 | 0.3379 | 0.6406 |
| LaBraM | **0.8002** | 0.8771 | 0.8810 | **0.4892** | **0.4336** | **0.7039** |
| BSR-VideoMAE (Joint train) | 0.7794 | 0.8652 | 0.8695 | 0.4206 | 0.4002 | 0.6862 |

## F    CHANNEL ABLATION VISUALIZATION

A visualization about the channel ablation experiment is shown in Figure 6.

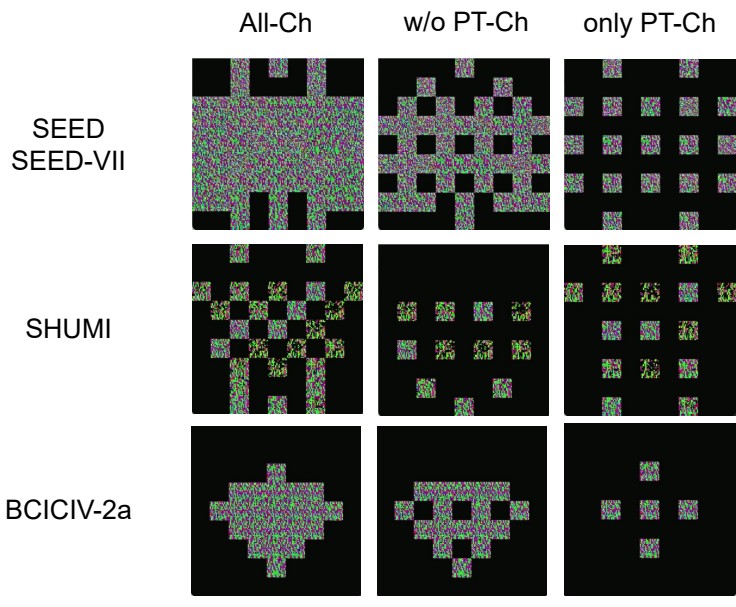

Figure 6: Channel ablation visualization.