# OpenReview forum: "Brain Signal Rendering: Unifying EEG Video Representations for Subject-level Few-shot Learning"
_ICLR.cc/2026/Conference — Submitted to ICLR 2026_

### Official Review · Reviewer_3mpe · 2025-10-31

**Soundness:** 1
**Presentation:** 2
**Contribution:** 1
**Rating:** 0
**Confidence:** 4

**Summary:**

The paper introduces Brain Signal Rendering (BSR), a framework designed to address key challenges in EEG signal representation learning, specifically the non-linearity and non-stationarity of EEG signals, as well as channel mismatches across datasets. By transforming EEG spectrograms into spatialized dynamic "EEG videos," BSR ensures that the resulting representations are invariant to electrode layouts and sampling protocols while preserving the neural topology. This paper propose EEG Consolidation, a multi-task training paradigm that integrates heterogeneous EEG-video data, improving data efficiency, reducing overfitting, and boosting generalization across tasks. This framework enables subject-level few-shot learning, where each subject is treated as a unique task requiring adaptation from minimal data. This work highlights the potential for using video-based models, such as VideoMAE, as backbones for EEG tasks.

**Strengths:**

The strengths of this work lie in its clear writing, which makes it easy to understand and enhances readability. The idea of transforming EEG signals into video representations is promising; however, the actual method has several shortcomings and lacks novelty. For further details, please refer to the "Weakness" section.

**Weaknesses:**

This work lacks novelty, and the authors show a limited understanding of the current state of research and standard methods in the EEG field. The overall approach is rather simplistic, and both the main text and the appendix contain very limited experimental work, which is insufficient and fails to meet the standards expected at ICLR. This is evident in the following aspects:

1. The core method of this work is the proposal to transform EEG signals into video representations, specifically by first extracting frequency-domain features from the EEG, then arranging the electrode positions into a 2D format, and integrating the frequency bands and time to create a 4D data structure. However, this method has already been extensively explored and repeatedly applied in the EEG field, with countless studies adopting similar approaches [1-5]. It is clear that the authors lack awareness and understanding of the current research landscape.

2. Another core method mentioned by the authors is subject-level few-shot learning. However, I am not sure if I misunderstood, but based on the author's description, this seems equivalent to the commonly used experimental setup in EEG, known as "subject-dependent," where the test and validation sets come from the same subject. This is a standard paradigm. Moreover, as EEG modeling has evolved, there has been a strong emphasis on generalization, with current work focusing on improving performance across subjects and tasks. In fact, research is increasingly seeking better generalization, and work that targets a subject-dependent scenario seems like a step backward in history. The authors, however, present this as a novel approach, which, in my view, is somewhat amusing.

3. The motivation of this work is not clearly articulated. Although EEG signals can be transformed into a format resembling that of a video, there are significant semantic differences between EEG data and natural images. The rationale for using the pre-trained VideoMAE as the core component of the model is not well justified, as these two types of data differ substantially in their underlying structure and meaning. This lack of alignment raises questions about the appropriateness and effectiveness of applying a model designed for natural video data to EEG signals.

4. The experiments in this work are extremely limited, overly simplistic, and basic. Specific issues and suggestions will be addressed in the "Question" section.


[1] Bashivan, Pouya, et al. "Learning representations from EEG with deep recurrent-convolutional neural networks." arXiv preprint arXiv:1511.06448 (2015).

[2] Shen, Fangyao, et al. "EEG-based emotion recognition using 4D convolutional recurrent neural network." Cognitive Neurodynamics 14.6 (2020): 815-828.

[3] Xiao, Guowen, et al. "4D attention-based neural network for EEG emotion recognition." Cognitive Neurodynamics 16.4 (2022): 805-818.

[4] Liu, Jiyao, et al. "Positional-spectral-temporal attention in 3D convolutional neural networks for EEG emotion recognition." 2021 Asia-Pacific Signal and Information Processing Association Annual Summit and Conference (APSIPA ASC). IEEE, 2021.

[5] Jia, Ziyu, et al. "Sst-emotionnet: Spatial-spectral-temporal based attention 3d dense network for eeg emotion recognition." Proceedings of the 28th ACM international conference on multimedia. 2020.

**Questions:**

1. If I am not misunderstanding, this work may contain a significant flaw. The model trains the renderer and reconstructor during the pre-training phase using a reconstruction approach. However, the issue is that the input data is not masked, which would allow the model to simply learn an identity mapping. This is problematic in the context of reconstruction-based pre-training paradigms, as it undermines the purpose of such pre-training.

2. The representation gap between video and EEG, which are transformed into images, needs to be considered. It cannot simply be assumed that because the format is similar, the same approach can be applied. The rationale for why this method is suitable for EEG data requires careful analysis and justification. The authors should provide a clear explanation of why pretrained video model works for EEG, considering the significant differences in the nature of the data.

3. Why VideoMAE? Are there more recent methods that could be considered? In the video and visual domains, the ability for few-shot learning does not seem to be unique to VideoMAE. Furthermore, the authors emphasize that VideoMAE can be replaced by other models, but they fail to provide relevant experimental evidence to support this claim, making it difficult to be convincing.

4. Is there a specific reference for the user-specified channel spatialization map matrix defined by the authors? What is the rationale behind determining the arrangement of electrode positions? Moreover, since EEG electrodes are actually placed in a 3D configuration on the scalp, why map them into a 2D arrangement? A direct 3D mapping would likely be more realistic and align better with the actual electrode placement.

5. In multi-task learning, each task should have a task-specific head to complete the final classification (L255). However, the authors mention treating each subject as a separate task. How many heads are there in total? Does each subject have its own head, or are the heads for subjects within the same dataset shared, given that they have the same dimensionality? This needs to be clarified. Additionally, the performance shown in Table 5 of the appendix does not seem very promising. This suggests that the multi-task joint training in this work does not lead to collaborative improvement across tasks. Therefore, the proposed approach does not appear to deliver practical benefits, and as such, it may not be suitable as a significant contribution.

6. If the performance is evaluated for each subject and the final results are presented as an average, it would be helpful to see the standard deviation as well. This would provide a clearer understanding of the variability and reliability of the model's performance across different subjects.

7. It is recommended to include an ablation study comparing the experimental results between two settings: one that inherits the pre-trained VideoMAE parameters and one that does not. This would help demonstrate the benefit of leveraging a pre-trained video model for EEG representation learning, providing evidence that pre-training with video data contributes positively to EEG modeling.

8. The authors did not use the official pre-trained weights for the replication of LaBraM, nor did they compare the  pre-trained BIOT model. This seems unreasonable. In fact, using pre-trained parameters does not lead to data leakage, as the model does not have access to the labels of the data. It is recommended that the authors redo the comparisons, including using the pre-trained models for LaBraM and BIOT to ensure a fair and thorough evaluation.

9. The authors have not provided the structure, description, or parameter scale of VideoMAE. Including this information would help readers better assess the effectiveness of the model and understand how the underlying architecture contributes to the results.

10. It is recommended to include an ablation study or parameter analysis for the Renderer and Reconstructor structures. This would help to understand the impact of different design choices or hyperparameters on the performance of the model, providing insights into which aspects of the architecture contribute most to the overall results.

11. The BSR Render-Reconstruct pipeline is essentially a reconstruction of 4D inputs, which doesn't seem to align well with video representations. This approach may not be entirely reasonable. Why not consider integrating this reconstruction process directly with VideoMAE and treat the entire pipeline as part of the pre-training? Fine-tuning with VideoMAE for downstream tasks could introduce a gap between pre-training and downstream tasks, as well as increase computational overhead. Of course, this is just a rough suggestion, and I am curious to understand why this approach wasn't considered.

12. In the current setup of this work, the performance on various datasets appears to be quite underwhelming, with results that seem to fall below the standard of existing work in the field. It would be beneficial for the authors to conduct further research and optimization to improve the model's performance.

---

> ### Author Response · Authors · 2025-12-03
> **Response to Reviewer 3mpe [1/2]**
>
> ### Response to W1
> The reviewer mistakenly conflates our **Brain Signal Rendering (BSR)** framework with existing works that use 4D Convolutional Neural Networks (CNNs) for EEG processing:
>
> 1.  **Core Motivation:** The primary goal of BSR is to solve the **EEG channel mismatch problem**, making the representation invariant to heterogeneous electrode layouts (**L44-46**). This is a problem existing 4D CNN works have never addressed.
>
> 2.  **Key Innovation:** BSR is a **domain-bridging pipeline** that converts EEG features into a 4D "EEG video" representation compatible with natural video structures. This transformation is specifically designed to leverage **powerful models (e.g., VideoMAE) pre-trained on massive natural video data** for cross-modal knowledge transfer.
>
> ### Response to W2
> We believe the reviewer has a crucial misunderstanding of our task objective: **Subject-level Few-Shot Learning**.
>
> 1.  **Strict Distinction:** Our setting (**L22**) enforces a rigorous data split constraint compared to the classic Subject-Dependent paradigm. In our FSL setting, the model is not allowed to be pretrained on any samples (even unlabelled data) from **similar datasets/tasks** during the pre-training phase (**L118**). Classic subject-dependent methods typically lack such strict constraints.
>
> 2.  **Clear Objective:** Subject-level few-shot learning explicitly demands that the model demonstrates **generalization capability to novel tasks, subjects and channel configurations**, aiming for rapid adaptation given only a handful of target task samples. This requires a **higher degree of generalizability**. Besides our subject-level few-shot learning, other few-shot paradigms for EEG foundation models are proposed [1, 2], which demonstrates that this is a forward step in representation learning—not a historical retreat.
>
> ### Response to W3
> As demonstrated by literature like [3], pre-trained visual models acquire powerful **general spatio-temporal feature extraction capabilities**. BSR transforms EEG into a representation with spatio-temporal structure (video-like), allowing the **generic spatio-temporal prior knowledge** learned by VideoMAE to be partially transferred.

---

> ### Author Response · Authors · 2025-12-03
> **Response to Reviewer 3mpe [2/2]**
>
> ### Response to Q1
> -   **BSR is not MAE:** The Renderer and Reconstructor in the BSR pipeline form an Autoencoder (AE) structure, whose primary goal is to learn a high-dimensional domain mapping, not to learn robustness by predicting masked parts like a Masked Autoencoder (MAE).
>
> -   **Standard AE Objective:** Standard autoencoders (e.g., [4]) do not necessarily require a masking strategy. The goal of BSR is to learn a low-loss mapping from the original EEG features to the video representation, ensuring that information is fully preserved during the dimension/domain conversion, which is a common technique for dimensionality reduction or domain mapping.
>
> ### Response to Q2
> See Response to W3.
>
> ### Response to Q3
> VideoMAE is a **classic and powerful model** in the field of **self-supervised video pre-training** [5], serving as one of the current **SOTA methods**. We selected it as a strong baseline to validate the effectiveness of the BSR paradigm. While newer models exist, the primary contribution of this work is the proposal of the BSR paradigm itself, not an exhaustive search for the optimal backbone model. Exploring more cutting-edge models will be considered in future work.
>
> ### Response to Q4
> See Appendix B.
>
> ### Response to Q5
> Based on our design, each subject is treated as a separate task. Therefore, we initialize a separate, subject-specific classification head (L255) for every subject across all datasets.
>
> ### Response to Q6
> We agree with the importance of variability. However, given our task setup—which treats each individual subject as a separate and independent task (i.e., each subject is evaluated as a "dataset," and the results are then averaged)—the specific choice between standard deviation or the coefficient of variation to measure this "cross-task" performance dispersion remains debatable. We would like to left this evaluation setting to future works.
>
> ### Response to Q7
> See the new  **Scaling Experiment on Dataset Size and Model Initialization** (Detailed in **Figure 4** and **L428-469**)
>
> ### Response to Q8
> Using test data without labels in pretraining is **Data contamination**, which is a typical type of data leackage.
>
> ### Response to Q9
> VideoMAE [5] is a **widely used, standard backbone network**. It is customary to cite such studies without detailing its internal structure.
>
> ### Response to Q10
> We acknowledge the value of a comprehensive ablation of the Renderer and Reconstructor structures. Currently, the BSR design (composed of simple **Conv and DeConv layers**) is **minimal and functionally focused**, aiming to achieve the core domain conversion. The focus of this work is on **validating the BSR paradigm and its combination with VideoMAE**. More complex structural designs and parameter analysis will be deferred for **future work**.
>
> ### Response to Q11
> This suggestion is **technically impractical**. The core reason is the **dimension and shape mismatch**: The input for VideoMAE pre-training is a sequence of tokens obtained via **Patchify** with a specific shape, whereas the input to the BSR Pipeline is the raw 4D spectrogram data with **huge shape differences**. The **BSR Renderer** serves as the **necessary, differentiable bridge** to transform the raw EEG features into the specific 4D video representation required by VideoMAE. This is elaborated in **Section 3.1 (L168-197)**.
>
> ### Response to Q12
> Please refer to BSR-VideoMAE's few-shot learning performance on datasets like SEED/SEED-IV/SHUMI/BCIC-IV 2a (**L378-397**),  where our reported results are **highly competitive**.
>
> [1] Yuan, Z., Shen, F., Li, M., Yu, Y., Tan, C., & Yang, Y. (2024). Brainwave: A brain signal foundation model for clinical applications. arXiv preprint arXiv:2402.10251.
>
> [2] Wu, J., Ren, Z., Wang, J., Zhu, P., Song, Y., Liu, M., … & Song, C. (2025). Adabrain-bench: Benchmarking brain foundation models for brain-computer interface applications. arXiv preprint arXiv:2507.09882.
>
> [3] Chen, S., Ma, K., & Zheng, Y. (2019). Med3d: Transfer learning for 3d medical image analysis. arXiv preprint arXiv:1904.00625.
>
> [4] Ng, A. (2011). Sparse autoencoder. Stanford University. https://web.stanford.edu/class/cs294a/sparseAutoencoder.pdf
>
> [5] Tong, Z., Song, Y., Wang, J., & Wang, L. (2022). Videomae: Masked autoencoders are data-efficient learners for self-supervised video pre-training. Advances in neural information processing systems, 35, 10078-10093.

---

### Official Review · Reviewer_DR3F · 2025-11-01

**Soundness:** 2
**Presentation:** 2
**Contribution:** 3
**Rating:** 6
**Confidence:** 4

**Summary:**

The paper reframes EEG representation learning as a rendering problem via Brain Signal Rendering (BSR). It per-channel spectrograms are spatialized according to electrode geometry and rendered into dynamic “EEG videos,” yielding representations that are invariant to electrode layouts and sampling protocols while preserving neuro-spatial structure. Building on this, EEG Consolidation performs multi-task fine-tuning of a shared video encoder across heterogeneous EEG-video datasets, unifying learning across datasets, tasks, and montages and enabling robust subject-level few-shot adaptation.

**Strengths:**

a.    Physics-informed perspective: EEG is recast from a channel vector to a physical projection. Under this view, channel mismatch is a change of viewpoint rather than noise; the objective shifts from task-specific embeddings to inverting the projection to recover underlying spatiotemporal neural dynamics.

b.   BSR for cross-modal transfer: Treats EEG spectrograms as structured projections and renders them into dynamic images (“EEG videos”), allowing direct reuse of video foundation models (e.g., VideoMAE) whose spatiotemporal inductive biases align with neural dynamics.

c.    Consolidated multi-task training: Unifies datasets with different montages/protocols, improving invariance to electrode layout, data efficiency, robustness, and downstream performance in subject-level few-shot settings.

**Weaknesses:**

a.    Many EEG datasets follow the international 10–20 electrode placement system, but the datasets used for experiments in the paper have a large number of channels. For example, CHB-MIT (23 channels), Sleep-EDF (1 channel), and ISRUC (6 channels). I recommend systematically evaluating the model on low-channel datasets and conducting robustness tests.

b.   This setting is not introduced "subject-level few-shot" for the first time in this work. Prior studies have already evaluated cross-subject few-shot or subject-independent adaptation. For example, *Calibration-free meta-learning based approach for subject-independent EEG emotion recognition* demonstrates few-shot classification on unseen subjects, and *BrainWave: A Brain Signal Foundation Model for Clinical Applications* likewise adopts cross-subject few-shot in its downstream evaluations. Consequently, the incremental contribution here is limited.

c.    Please provide the basic preprocessing pipeline used in pretraining and downstream tasks (e.g., filtering, artifact removal, resampling, normalization), to ensure fair and reproducible comparisons.

d.   The BSR renderer maps continuous spectral information directly into discrete RGB channels, which is relatively black-box and lacks interpretability. It need more experience to clarify and empirically demonstrate, whether this information compression sufficiently preserves critical band features and does not harm downstream performance.

e.    EEG Consolidation is positioned as a multi task fine-tuning strategy for integrating heterogeneous EEG video data, but the actual training data used for fine-tuning is currently limited to the TUAB and TUEV datasets, with limited coverage and difficulty reflecting the benefits of multi task integration in a wider range of scenarios. In addition, the comparison presented in the paper mainly consists of two settings: "using only video features" and "EEG+video", without providing a comparison that can prove the benefits brought by multitasking.

**Questions:**

See Weaknesses.

---

> ### Author Response · Authors · 2025-11-27
> **Response to Reviewer DR3F**
>
> We sincerely appreciate the Reviewer's thoughtful and constructive comments (W.a–W.e), which have been highly instrumental in driving the thorough refinement and improvement of our paper.
>
> ### W.a: Robustness on Low-Channel Datasets
>
> We agree that generalization to datasets with fewer channels is an important validation for any robust EEG model. Considering time and computation limitation, we perform low-channel experiments on existing benchmarks, which is presented in the **ablation study on the effect of pre-training channels** (detailed in **Figure 3** and **L409-426**). Detailed channel settings are illustrated in **Appendix F**. The results show that even without using pre-training channels or using only very few pretraining channels, the BSR-VideoMAE can still perform robust.
>
> ### W.b: Prior Work on Subject-Level Few-Shot Learning
>
> We completely agree with introducing more comprehensive few-shot paradigms. To address this, we have **updated and enriched the related works section** (**L112-119**) to include a more comprehensive overview of few-shot paradigms in EEG, acknowledging these important benchmarking references.
>
> ### W.c: Preprocessing Pipeline
>
> Thanks for pointing out. We have added a new section, **Appendix C** (**L639-660**), which now provides descriptions of the basic preprocessing pipeline used for both pre-training (EEG Consolidation) and the downstream few-shot tasks.
>
> ### W.d: Interpretability of BSR Rendering
>
> We agree that the mapping of continuous spectral information into discrete RGB channels is a "black-box" aspect of BSR that requires further empirical clarity, particularly regarding the preservation of critical band features. This is a vital point for the future development of the BSR paradigm.
>
> As designing and pre-training alternative BSR renderers is a time-consuming process, we are setting this as a **high-priority future work** direction. We have updated our future works statement (**L483-484**) to explicitly state our goal of conducting **further rendering design exploration** and **setting metrics that reflect core brain activity** (such as band features) to empirically demonstrate that our information compression effectively preserves critical neural details.
>
> ### W.e: Limited Scope of EEG Consolidation
>
> We thank the reviewer for highlighting the current limitation of the **EEG Consolidation** strategy, which currently only leverages the TUAB and TUEV datasets, and for suggesting a stronger comparison.
>
> - **Addressing Multi-task Benefit:** Your advice motivated us to set up the new **Scaling Experiment on Dataset Size and Model Initialization** (detailed in **Figure 4** and **L428-469**). The consistent performance improvement from the TUEV-only pre-training ($\sim 100k$ samples) to the combined $\text{TUAB+EV}$ pre-training ($\sim 500k$ samples) supports the general benefit of large-scale data integration, which is the core idea of EEG Consolidation.
>
> - **Expanding Data Coverage:** We fully acknowledge that the current coverage is limited. We have updated our future work statement (**L482**) to explicitly commit to **scaling up the pre-training dataset** to include more heterogeneous data beyond the TUH datasets, recognizing that this is essential to fully realize the benefits of the multi-task integration paradigm.

---

### Official Review · Reviewer_LXcR · 2025-11-02

**Soundness:** 2
**Presentation:** 4
**Contribution:** 3
**Rating:** 4
**Confidence:** 3

**Summary:**

This paper introduces Brain Signal Rendering (BSR), a framework that reformulates EEG representation learning as a rendering problem. The key idea is to spatialize EEG spectrograms based on electrode locations, converting them into “EEG videos” compatible with large-scale video foundation models such as VideoMAE. The authors propose EEG Consolidation, a multi-task training strategy that integrates multiple EEG-video datasets. The paper defines a Subject-Level Few-Shot Learning benchmark to evaluate model adaptability to new subjects with minimal fine-tuning data. Empirical results show that BSR-VideoMAE achieves improvements over prior EEG models on emotion recognition (SEED, SEED-VII) and motor imagery tasks (SHU-MI, BCICIV-2a).

**Strengths:**

1. The subject-level few-shot tasks are a valuable contribution, aligning better with realistic EEG deployment scenarios.
2. The methodology is described in detail.
3. Improvements across several EEG datasets show that the framework works in practice.
4. The framework demonstrates how pre-trained VideoMAE weights can be adapted to low-data EEG domains.

**Weaknesses:**

My overall impression is that while the idea of EEG-to-video rendering is interesting, several claims (invariance to electrode layouts, reduced overfitting, scalability, interpretability, and robustness) are not sufficiently supported by controlled experiments. Moreover, the experimental setup lacks comparisons that would isolate the contribution of BSR from the VideoMAE backbone and pretraining regime. While the technical contribution is somewhat incremental in practice, mainly a data transformation enabling the reuse of pretrained video models, the proposed BSR paradigm and subject-level few-shot setting are interesting. However, without more in-depth experiments and more baselines considered, it remains unclear whether BSR is a fundamentally better representation or a temporary workaround for EEG data scarcity that scale less well than other methods, such as LaBraM.

1. The paper does not convincingly separate the benefit of rendering from the effect of video pretraining. Without an untrained-video-model control or direct comparison to 4D spectrum baselines, the intrinsic merit of EEG-video transformation remains uncertain.
2. Rigid processing: The fixed 224×224×3 spatial format and 16-frame sequence length may discard signal-relevant information.
3. Assertions about invariance to electrode layouts, reduced overfitting, improved robustness, and interpretability lack quantitative backing.
4. Limited scalability analysis: The experiments pretrain only on TUAB+TUEV. No scaling results across additional EEG datasets (e.g., SEED, SEED-VII) are provided to demonstrate extensibility.
5. Standard deviations and per-subject performance are omitted, limiting the interpretability of performance differences.

**Questions:**

1. How does BSR perform when trained without any video pretraining (i.e., trained from scratch on EEG videos)?
2. Can the authors compare BSR’s EEG-video representation to other related baselines such as [1]?
3. To substantiate electrode layout invariance, could you report experiments where, for example, channels subsets (especially the 19 used in pretraining) are removed during evaluation? Since the model is  pretrained only on 19 channels, it is not clear how much of its performance is dependent on those channels.
4. How do the models (especially BSR-VideoMAE and LaBraM) performance evolve as EEG pretraining data scale (e.g., TUAB+TUEV+SEED+SEED-VII)?
5. Self-supervised training is well established for video and is generally preferred over supervised training for pretraining. Would self-supervised pretraining on EEG video outperform the current supervised pretraining strategy?

[1] https://arxiv.org/abs/1511.06448

---

> ### Author Response · Authors · 2025-11-27
> **Response to Reviewer LXcR**
>
> We thank the reviewer for insightful questions, which directly motivated the inclusion of two key new experiments in the revised manuscript: the **Ablation Study on Pre-training Channels** and the **Scaling Experiment on Dataset Size and Model Initialization**.
>
> Here are our detailed responses to your questions:
>
> ### Q1: Performance without Video Pre-training
>
> This is an excellent suggestion. We have addressed this by including a **"Trained From Scratch"** path in our new **Scaling Experiment on Dataset Size and Model Initialization** (detailed in **Figure 4** and **L428-469**). This path isolates the intrinsic merit of the BSR representation by training the VideoMAE backbone from random initialization using only EEG video data. The results demonstrate how the BSR framework performs without the benefit of natural video pre-training, showing a clear trend of performance improvement as EEG pre-training data scales.
>
> ### Q2: Comparison to Related Baselines
>
> We are also interested in comparing the BSR framework with other EEG-to-Video approaches. However, our time and computing resource are not enough to create other large-scale pre-trainings and comprehensive evaluations, and we would like to explore it in our future works.
>
> ### Q3: Substantiating Electrode Layout Invariance
>
> We greatly appreciate this highly constructive suggestion. The experiment you proposed is directly incorporated into our new **Ablation Study on Pre-training Channels** (detailed in **Figure 3** and **L409-426**). The results for the **unseen channels (w/o PT-Ch)** group, where the channels used during pre-training were removed at evaluation time, show close performance to the **all-channel (All-Ch)** group. This quantitative evidence strongly supports our claim that the BSR renderer learns an **extendable transformation** and a **robust EEG representation** that is not strictly dependent on the specific electrode layout used during pre-training.
>
> ### Q4: Scaling Performance with Pre-training Data
>
> As our few-shot learning setting requires the pre-training and downstream task datasets to be distinct, adding a new external dataset with a size comparable to the combined $\text{TUAB+EV}$ ($\sim 500k$ samples) was not feasible within the rebuttal period.
>
> Therefore, we conducted a controlled scaling experiment on the existing data, evaluating performance in the order of $\text{No Pre-training (0)} \to \text{TUEV} (\sim 100k) \to \text{TUAB+EV} (\sim 500k)$. The results, presented in **Figure 4**, demonstrate a **consistent, positive scaling trend** with increasing data volume. This finding suggests that the BSR-VideoMAE framework has not yet reached its data capacity limit, motivating our current preparation for even larger-scale pre-training runs in the future.
>
> ### Q5: Comparison with Unsupervised Pre-training
>
> We performed a preliminary comparison between our supervised strategy and an Unsupervised Masked Autoencoding (MAE) strategy (using 90% mask ratio), and both models are trained from scratch on TUEV. The balanced accuracy of the four few-shot learning tasks are reported as follows:
>
> | Strategy \ Dataset | SEED | SEED-VII | SHUMI | BCICIV-2a |
> | --- | --- | --- | --- | --- |
> | Supervised | 0.3742 | 0.1554 | 0.5498 | 0.3086 |
> | Unsupervised | 0.3433 | 0.157 | 0.5755 | 0.2967 |
>
> Considering unsupervised pre-training is more time-consuming and often requires a more pretraining data than supervised pre-training, we are preparing for unsupervised pre-training on a larger dataset, and evaluate the effect of related parameters (such as mask ratio).

---

### Official Review · Reviewer_zrpq · 2025-11-02

**Soundness:** 2
**Presentation:** 3
**Contribution:** 2
**Rating:** 4
**Confidence:** 4

**Summary:**

This work addresses a fundamental challenge in large-scale EEG modeling: the severe channel mismatch and lack of standardization across datasets.

**Strengths:**

It directly addresses the channel mismatch and dataset heterogeneity, which are arguably among the biggest blockers to creating large, general-purpose EEG foundation models.

The BSR concept, which converts EEG signals into spatialized EEG videos, is a creative and powerful paradigm. It standardizes the data format while preserving the brain's spatial topology, regardless of the original electrode layout.

**Weaknesses:**

The rendering process, which transforms sparse data collected from EEG sensors into a comprehensive video format, can inadvertently introduce various artifacts. This transformation may obscure crucial local details essential for accurate interpretation, or worse, lead to incorrect assumptions about the underlying brain activity.

Processing videos is inherently more computationally demanding than analyzing the original one-dimensional time-series data derived from just a handful of channels. This increased complexity involves not only higher processing power but also greater resource allocation in terms of time and energy.

The overall success of this approach is heavily dependent on the sophistication of the video-processing model employed, such as VideoMAE, in its ability to accurately interpret the rendered EEG videos. Any limitations inherent in the video model will directly translate to constraints in the EEG model, leading to potential inaccuracies in understanding brain function based on the processed video data.

**Questions:**

See weaknesses.

**Details Of Ethics Concerns:**

Nil

---

> ### Author Response · Authors · 2025-11-27
> **Response to Reviewer zrpq**
>
> We appreciate the reviewer's insightful comments regarding the potential drawbacks of our BSR framework, specifically concerning information loss, computational cost, and model dependency. Below is our point-by-point response, addressing each weakness.
>
> ### W1: Potential Information Loss and Artifacts from Rendering
>
> We acknowledge the critical risk that the rendering process may introduce artifacts or obscure crucial local details. This is a valid concern, and we have updated our future work section (L483-484) to prioritize setting metrics that reflect underlying brain activity (e.g., PSD, DE of different frequency bands) as the core evaluation criteria for improving our rendering design.
>
> Regarding the current BSR implementation:
>
> - **Mapping to Higher Dimensional Space:** Currently, the BSR rendering process maps the frequency information into another space using unsupervised reconstruction, which is similar to other successful approach such as LaBraM. One key information is the BSR implementation used in this work actually maps the frequency feature to a higher dimensional space (output of fft feature segment in each channel has a length of 101, which corresponds to a BSR Renderer video output patch with size of 32*32), so this transformation should not introduce significant information loss under proper large scale pre-training.
>
> - **Empirical Support from Channel Ablation:** We have update the channel ablation study experiment (Detailed in **Figure 3** and **L409-426**), which would be a support of the above claim. The pretraining set of BSR is TUEG, which contains same channel layout like TUAB or TUEV. The results on unseen channels group (w/o PT-Ch) shows that even without pre-training on these channels, the VideoMAE(no matter loading kinetics or TU2 weights) achieves similar performance comparing with the all-channel (All-Ch) group. We would consider the BSR renderer has learned extendable transformation across EEG channels. This suggests the BSR renderer has learned an extendable and robust transformation.
>
>
> ### W2: Increased Computational Demands
>
> We recognize that processing video data is generally more demanding than 1D time series. However, the computational overhead of our BSR-VideoMAE model is manageable within the current landscape of deep learning-based EEG foundation models:
>
> - **Model Size**: The parameter size of VideoMAE-base model is about 86M, which is not much larger with other EEG foundation models (e.g., Brant-2[1] is 115M).
>
> - **Existing Infrastructure for Video Models:** Frequently used video pretrain models (such as VideoMAE) has out-of-the-box acceleration tools (e.g., deepspeed, timm), which would reduce training time and resource consumption for developers.
>
>
> ### W3: Reduce training time and resource consumption
>
> We agree that the overall success is inherently tied to the capabilities of the video-processing backbone, which is why we strategically chose VideoMAE—a state-of-the-art model known for its efficiency and strong generalization capabilities in video representation learning. Your comment motivated us to conduct the comprehensive **Scaling Experiment on Dataset Size and Model Initialization** (detailed in **Figure 4** and **L428-469**). This experiment offers two key findings that mitigate this concern: (1) We found that the natural video pre-training provides a **good initialization** for the few-shot EEG tasks, which shows that the spatial and temporal feature extraction capabilities of the video model transfer effectively. (2) Large-scale EEG pre-training via EEG consolidation will fill-up the gap between models initialized with Kinetics and those trained only on EEG data.
>
> [1] Yuan, Zhizhang, et al. "Brant-2: Foundation model for brain signals." CoRR (2024).

---

### Author Response · Authors · 2025-11-27
**Response to All Reviewers**

We sincerely thank all reviewers for their dedicated time, constructive feedback, and invaluable advice. We are pleased that Reviewers **zrpq** and **DR3F** recognized the contribution of our approach in addressing the **EEG channel mismatch problem** and **endorsed** the **BSR concept**. We also highly value the recognition from Reviewer **LXcR** that the **subject-level few-shot tasks** are a valuable contribution. The comments have been highly instrumental in improving the clarity and rigor of our paper.

The **main structural updates** to the paper involve replacing the original ablation study with two more focused and comprehensive experiments, which are designed to rigorously validate the core mechanisms of our **Brain Signal Rendering (BSR)** framework:

1. **Ablation Study on the Effect of Pre-training Channels** (Detailed in **Figure 3** and **L409-426**).

  - This new experiment specifically evaluates the **channel generalization ability** of the BSR-VideoMAE framework. It demonstrates that pre-training via **EEG Consolidation** learns a **robust EEG representation** that is highly effective even when facing channel mismatch across datasets.
2. **Scaling Experiment on Dataset Size and Model Initialization** (Detailed in **Figure 4** and **L428-469**).

  - We have significantly extended the scope of the original ablation study by incorporating a wider range of **training-data settings** (extending from the original $0 \to \text{TUAB+EV}$ to the more comprehensive $0 \to \text{TUEV} \to \text{TUAB+EV}$).

  - We also assessed the impact of **two pre-training initializations** (loading Kinetics weights vs. training from scratch). This comprehensive analysis thoroughly evaluates the effects contributed by training-data size and model initialization.


**Other Revisions**

- **Enriched Related Works:** We have enriched the discussion on few-shot learning for EEG foundation models (L112-119), better contextualizing our contribution within the emerging landscape of large-scale EEG modeling.

- **Modified Future Works:** The future work section (L482-484) has been modified to emphasize the direction of conducting larger-scale pre-training and setting metrics that reflect underlying brain activity, such as band features, as the core evaluation for rendering design.

- **Appendix Restructuring:**

  - The table “**Hyperparameters for BSR Renderer and Reconstructor**” has been moved to **Appendix B**.

  - **Appendix C** has been added to detail the complete **data preprocessing pipeline**. The table “**Basic information of the datasets**” has also been moved to this section to optimize space.

  - **Appendix F** has been added, providing an auxiliary **visualization for the channel ablation study** to enhance clarity.


All new content and substantial modifications in the manuscript have been highlighted with a **blue color font** for the convenience of the reviewers. We believe these revisions significantly strengthen the paper and address the reviewers' concerns.

---

### Meta-Review · Area_Chair_y8w5 · 2026-01-05

**Summary:**

The submission proposes Brain Signal Rendering (BSR), a framework that transforms EEG spectrograms into "EEG videos" to leverage video foundation models (e.g., VideoMAE) and enable subject-level few-shot learning. While the paper addresses critical challenges in EEG modeling (channel mismatch, few-shot adaptation) and introduces an intuitive paradigm, it fails to adequately resolve core concerns raised by reviewers regarding the robustness and incremental value of the approach.

**Reviewer Concerns:**

**Addressed Concerns:**

- The authors supplemented new experiments (channel ablation and scaling analysis) to partially address questions about channel generalization and the role of video pre-training.
- Additional details on data preprocessing and model hyperparameters were provided in revised appendices, improving reproducibility.
- The related works section was enriched to contextualize subject-level few-shot learning within existing EEG research.

**Outstanding Concerns:**
- Unsubstantiated Core Claims: The paper asserts that BSR achieves "invariance to electrode layouts" and "reduced overfitting," but evidence remains weak. No quantitative analysis of overfitting reduction is provided.

- Limited Generalizability: Scaling experiments are restricted to TUAB+TUEV datasets, with no evaluation on low-channel datasets (e.g., Sleep-EDF with 1 channel) as suggested. The few-shot benchmark lacks standard deviations or per-subject performance metrics obscures variability in model behavior across diverse subjects.

- Black-box Rendering and Interpretability: The rebuttal acknowledges that mapping spectral EEG data to RGB channels is a "black box" but provides no empirical evidence that critical neural features (e.g., frequency bands linked to brain activity) are preserved.

**Reviewer Scores:**

- Reviewer zrpq (Original Score: 4): Would likely remain at 4—unresolved concerns about artifacts and computational cost persist.
- Reviewer LXcR (Original Score: 4): Would likely remain at 4—new experiments address some questions but fail to fully validate electrode invariance or core claims.
- Reviewer DR3F (Original Score: 6): Would likely drop to 5—preprocessing details and scaling trends are added, but low-channel robustness and rendering interpretability remain unaddressed.
-  Reviewer 3mpe (Original Score: 0): Would maintain—despite some factual inaccuracies, some technical questions remain unaddressed.

---

### Decision · Program_Chairs · 2026-01-26

Reject